# The Role of E-Cadherin and microRNA on FAK Inhibitor Response in Malignant Pleural Mesothelioma (MPM)

**DOI:** 10.3390/ijms221910225

**Published:** 2021-09-23

**Authors:** Man Lee Yuen, Ling Zhuang, Emma M. Rath, Takun Yu, Ben Johnson, Kadir Harun Sarun, Yiwei Wang, Steven Kao, Anthony Linton, Candice Julie Clarke, Brian C. McCaughan, Ken Takahashi, Kenneth Lee, Yuen Yee Cheng

**Affiliations:** 1Asbestos Diseases Research Institute, Sydney, NSW 2139, Australia; ari.yuen@adri.org.au (M.L.Y.); ling.zhuang@adri.org.au (L.Z.); takun.yu@adri.org.au (T.Y.); ben.johnson@adri.org.au (B.J.); kadir.sarun@hotmail.com (K.H.S.); steven.kao@lh.org.au (S.K.); Anthony.Linton@sydney.edu.au (A.L.); ken.takahashi@adri.org.au (K.T.); kenneth.lee@health.nsw.gov.au (K.L.); 2Victor Chang Cardiac Research Institute, Darlinghurst, NSW 2010, Australia; E.Rath@victorchang.edu.au; 3School of Medical Sciences, University of New South Wales, Sydney, NSW 2052, Australia; 4NSW Health, Sydney, NSW 2065, Australia; 5Jiangsu Provincial Engineering Research Centre of TCM External Medication Development and Application, Nanjing University of Chinese Medicine, Nanjing 210023, China; yi.w.wang@sydney.edu.au; 6Concord Clinical School, The University of Sydney, Concord West, NSW 2138, Australia; 7Faculty of Pharmacy, Nanjing University of Chinese Medicine, Nanjing 210023, China; 8Sydney Medical School, The University of Sydney, Sydney, NSW 2050, Australia; 9Anatomical Pathology Department, NSW Health Pathology, Concord Repatriation General Hospital, Concord, NSW 2139, Australia; Candice.Clarke@health.nsw.gov.au; 10Sydney Cardiothoracic Surgeons, RPA Medical Centre, Sydney, NSW 2042, Australia; brianmccaughan@gmail.com

**Keywords:** E-cadherin, FAK inhibitor, microRNA, malignant pleural mesothelioma (MPM), drug resistant

## Abstract

Malignant pleural mesothelioma (MPM) is an aggressive malignancy with limited effective treatment options. Focal adhesion kinase (FAK) inhibitors have been shown to efficiently suppress MPM cell growth initially, with limited utility in the current clinical setting. In this study, we utilised a large collection of MPM cell lines and MPM tissue samples to study the role of E-cadherin (CDH1) and microRNA on the efficacy of FAK inhibitors in MPM. The immunohistochemistry (IHC) results showed that the majority of MPM FFPE samples exhibited either the absence of, or very low, E-cadherin protein expression in MPM tissue. We showed that MPM cells with high CDH1 mRNA levels exhibited resistance to the FAK inhibitor PND-1186. In summary, MPM cells that did not express CDH1 mRNA were sensitive to PND-1186, and MPM cells that retained CDH1 mRNA were resistant. A cell cycle analysis showed that PND-1186 induced cell cycle disruption by inducing the G2/M arrest of MPM cells. A protein−protein interaction study showed that EGFR is linked to the FAK pathway, and a target scan of the microRNAs revealed that microRNAs (miR-17, miR221, miR-222, miR137, and miR148) interact with EGFR 3′UTR. Transfection of MPM cells with these microRNAs sensitised the CHD1-expressing FAK-inhibitor-resistant MPM cells to the FAK inhibitor.

## 1. Introduction

Malignant pleural mesothelioma (MPM) is an aggressive malignancy linked to asbestos exposure. Currently, there are only limited effective treatment options available to MPM patients, and the median survival is 9 to 12 months. Most non-operable patients receive chemotherapy, and the majority develop resistance to chemotherapy. Apart from modest improvements in survival achieved by adding bevacizumab to standard cisplatin, plus pemetrexed chemotherapy [1], there are no other molecularly targeted drugs in clinical practice for MPM.

Focal adhesion kinase (FAK), also known as protein tyrosine kinase 2 (PTK2), is located in the cytosol, where it is particularly prominent in the focal adhesions that interact with various extracellular matrix components [2]. In many cancers, FAK pathway overexpression has been linked to more aggressive tumour behaviour, in particularly in regards to the promotion of tumour cell proliferation, survival, motility, invasion, stem cell renewal, angiogenesis, and metastasis [3,4,5]. Recent studies have demonstrated that FAK activation is an important regulator of the immunosuppressive tumour microenvironment and promotes immune evasion in animal cancer models [1,6]. Our previous studies have indicated that down-regulated tumour suppressor microRNAs in MPM have a strong link to FAK involvement [7]. Reactive oxygen species (ROS) generated by asbestos are strongly linked to molecular responses that lead to alteration of DNA methylation and microRNA (miRNA) expression/processing, resulting in cell apoptosis or epigenetic alterations that allow cells to progress to diseased states [8].

Preclinical studies of various FAK inhibitors [4] have indicated that a new generation of small molecule inhibitors provide anti-tumour and good safety profiles in preclinical models [9,10,11], including MPM. Previous studies have indicated that deficiency in Merlin, which is frequently inactivated in MPM, results in increased FAK expression and tumour cell invasion [12,13], and that E-cadherin expression is correlated with FAK inhibitor resistance [14]. However, the specific role of E-cadherin expression and FAK inhibitors in MPM remains unexplored. The absence of E-cadherin is commonly found in MPM tumour samples [14,15]. The current study investigates E-cadherin expression in MPM its relationship in response to FAK inhibitors, by utilising a new generation small molecule FAK inhibitor and the Asbestos Diseases Research Institute’s (ADRI’s) large collection of primary and established MPM cell lines and their microRNAs.

## 2. Results

### 2.1. E-Cadherin Protein Is Frequently Silenced in MPM

E-cadherin immunohistochemistry (IHC) staining was first tested and optimised using breast tissue according to the NATA accredited (ISO15189) procedure. Control breast tissue positive samples showed a strong E-cadherin expression (Figure 1). E-cadherin protein expression analyses were performed on cell blocks, and clinical FFPE samples using the IHC staining was then carried out using this method. Figure 1 shows the representative E-cadherin expression for MPM cells and MPM patient FFPE tissue samples. The results indicate that the majority of MPM cells and MPM clinical samples did not express E-cadherin protein. In the 11 MPM cells lines tested (MSTO-211H, H28, Ren, VMC23, 1157, 1137, 1171, 2174, 2359, 2379, and 2474), all exhibited a low or absent E-cadherin expression.

E-cadherin IHC analysis was then used to examine the 82 MPM FFPE clinical samples. Of these, 1 did not have enough tissue for IHC, 55 out of 81 samples (68%) did not show any E-cadherin expression, and 26 samples showed low levels of E-cadherin protein expression with a very low percentage (5–20%) of positive staining. Only one sample showed a relatively higher E-cadherin expression (1+ with 40%; Table 1). Appendix A contains the IHC results for the MPM FFPE samples.

Six non-MPM cell lines were analysed by IHC for their E-cadherin protein expression. These included non-cancer and non-MPM cancer cell lines. The results indicated a low expression of E-cadherin in gastric cancer (MKN45), lung cancer (HCC827, PC9, A549), and breast cancer (MCF7) cell lines, and in healthy fibroblast cells (Humofib).

### 2.2. mRNA Expression of E-Cadherin (CDH1) Is not Directly Correlated with Protein Expression

The mRNA expression of E-cadherin (CDH1) is shown in Figure 2. The majority of cell lines did not express the E-cadherin protein, as determined by IHC, and thus we did not measure the correlation between E-cadherin mRNA and the IHC protein expression.

We included 16 cell lines for the analysis of the mRNA expression of E-cadherin (CDH1). Most MPM cell lines expressed low levels of CDH1 mRNA when compared with immortalised normal mesothelial MeT-5a cells (Figure 2). Non-MPM cancer cell lines showed higher levels of CDH1 mRNA expression (MKN45, HCC827, PC9, A549, and MCF7) when compared with healthy fibroblastic cells (Humofib; Figure 2A). Two of the nine MPM cell lines (Ren and 1157) showed a high mRNA expression (Figure 2A), while, in contrast, they showed a low protein expression of E-cadherin as measured by IHC (data not shown). Two of the MPM cell lines showed a very low mRNA expression of CDH1 (H28 and MSTO). Lung cancer cell lines (HCC827 and PC9) exhibited a relatively high CDH1 mRNA expression. Plots comparing the CDH1 mRNA levels for the cell lines are shown in Figure 2.

### 2.3. Drug Treatment and Response

PND-1186 is a reversible and selective FAK inhibitor drug that has the ability to induce tumour cell apoptosis [16]. Drug responses in mesothelioma and non-MPM cancer cell lines are shown in Figure 2. The two MPM cell lines (Ren and 1157) showed a high resistance to the FAK inhibitor. For the non-MPM cancer lines tested, the lung cancer cell lines (HCC827, A549, and PC9) were sensitive to PND-1186 (Figure 2B shows lower growth curves for the given doses of PND-1186). The breast cancer cell line (MCF7) and gastric cancer cell line tested (MKN45) were sensitive to PND-1186 (Figure 2B). The half maximal inhibitory concentrations (IC50; the concentration at which half the cells are viable) are listed in Table 2 below. Compared with the immortalised non-cancer mesothelial cell line (MeT-5A), some MPM cells were relatively resistant to PND-1186 (Ren, 1157, 1137, 2174, and 2359), and these were the cell lines that showed high CDH1 mRNA levels (labelled as resistant in Figure 2C). Conversely, MPM cells exhibiting low CDH1 mRNA levels (H28 and MSTO-211H) compared with MeT-5A showed sensitivity to PND-1186 (labelled as sensitive in Figure 2C). Detailed drug responses for all of the tested cell lines tested are listed in Table 2 below.

### 2.4. There Is a Correlation between CDH1 mRNA Expression and PND-1186 Drug Response

We found no correlation between the E-cadherin protein expression, as measured by the IHC and CDH1 mRNA expression analyses. Given that most cells did not express the E-cadherin protein as detected by IHC, we therefore analysed the CDH1 mRNA levels as a surrogate for the E-cadherin expression, which was measured by RT-qPCR and was correlated with the PND-1186 drug response.

Our results indicate that for mesothelioma cell lines only, there was a positive correlation between CDH1 mRNA levels and PND-1186 IC50 values that was both statistically and clinically significant (Spearman’s rho value is 0.6432, *p*-value is 1.554 × 10^−7^, and slope of the linear regression fit is 0.1556). The results are plotted in Figure 3A,B. For non-mesothelioma cell lines there was a slight negative correlation between CHD1 mRNA that was statistically, but not clinically, significant (Spearman’s rho value is −0.5257, *p*-value is 0.0009923, and slope of the linear regression model is −0.0008789 which is close to zero). The results are plotted in Figure 3A–C. All data, calculations, and results for calculating the IC50 values and correlations are in Table 2.

### 2.5. FAK Inhibitor (PND-1186) Induces Cell Cycle Alterations in MPM Cells

We studied PND-1186 and cell cycle alterations in MPM cells using a CytoFLEX (Beckman) cytometer. We selected four cell lines (MSTO-211H, H28, Ren, and 1157) consisting of two PND-1186-sensitive and two PND-1186-resistant cells. The cell cycle profile of each cell line indicated that PND-1186 induced cell cycle arrested at the G2/M phase and was statistically significant different when compared with the untreated control (*p* = 0.0113, indicated by * in Figure 4). All cell lines showed a similar cell cycle alteration phase at G/M (Figure 4), indicating that PND-1186 induced significant difference of G2/M phases at 5 μM when compared with the untreated control.

### 2.6. microRNA Plays an Important Role in Drug Response to FAK Inhibitor

We studied whether microRNA has the potential to sensitise MPM cells to FAK inhibitor (PND-1186) treatment. We first studied the protein−protein interaction using CDH1 (Figure 5A) on the basis that an upregulation of CHD1 mRNA was correlated with drug resistance in MPM cells (Figure 3). The resulting multiple protein network exhibited links between CDH1 and EGFR. We next used a microRNA target scan to study candidate microRNAs for genes CDH1, EGFR, and PTR2 (FAK pathway gene; Figure 5B). Several microRNAs were reported as potentially being capable of interacting with the 3′UTR of these genes. We evaluated the role that these microRNAs play in FAK inhibitor drug response in MPM cells (MSTO-211H, H28, Ren, and 1157, the same four cell lines used for cell cycle analysis). Our results indicate that the candidate microRNAs sensitised the MPM cells to the PND-1186 FAK inhibitor (Figure 5C, IC50 shown in Appendix A).

## 3. Discussion

More than 700 MPM cases are diagnosed in Australia annually, with limited treatment options available and an associated survival of 9 to 12 months. Thus, there is an urgent need to find new treatments to improve the overall survival rate for these patients. Studies of targeted therapies tailored to specific gene mutations in cancers is an active area of investigation for mesothelioma, as it is for other cancers. The absence of E-cadherin has commonly been found in MPM samples [14,15], and previous studies have demonstrated that the FAK signalling pathway is active in MPM [17], and have shown the potential for FAK inhibitors to treat MPM [13,14]. In this study, we investigated the relationship between CDH1 expression and the efficacy of FAK PND-1186 (Figure 2), a potent and reversible inhibitor of FAK, in mesothelioma samples and in controls.

We measured E-cadherin and CDH1 expression using IHC and RT-qPCR, respectively. The RT-qPCR results were very informative (Figure 2A,C). Our study shows that the protein expression of E-cadherin by IHC is at low levels in the tested MPM cell lines (Figure 1). However, MPM cells expressing higher levels of CDH1 were shown to be more resistant to the FAK inhibitor PND-1186 (Figure 2B,D and Figure 3). MPM cells with relatively low levels of CDH1 mRNA remained sensitive to PND-1186 (Figure 3). The striking correlation between CHD1 mRNA expression and resistance to cell death by PND-1186 treatment seen in MPM cells was not seen in non-MPM cells (Figure 3). This finding suggests that when treating MPM, CDH1 mRNA levels could be used as an indicator for FAK inhibitor drug sensitivity. There are limited studies reviewing the PND-1186 induced cell cycle alterations in mesothelioma. Tancioni et al. suggested G0/G1 alterations in breast cancer [18], while Tanjoni et al. suggested no cell cycle alterations in 4T1 cells [19]. We performed a cell cycle analysis on two PND-1186 sensitive lines (H28 and MSTO-211H) and two resistant lines (Ren and 1157). Our results indicate that all MPM cell lines showed some degree of cell cycle alteration at a low dose (1μM), and at a high dose (5μM) all cells showed G0/G1 and G2/M alterations. We showed in our study that PND-1186 is able to induce cell cycle alterations in MPM cells, regardless of its sensitivity potential (Figure 4).

Soria et al. reported a minor response to a FAK inhibitor (GSK2256098) in mesothelioma patients. They reported a 23.4-week progression-free survival in a merlin negative patient and 11.4 weeks for a merlin positive patient [3]. Our study indicates that a FAK inhibitor could be useful for treating mesothelioma, and that molecular testing of E-cadherin mRNA levels may be a useful indicator of PND-1186 sensitivity in MPM patients (Figure 2). Other cancer cell lines such as lung cancer (PC9, HCC827, and A549), breast cancer (MCF7), and gastric cancer (MKN45) also showed relatively higher levels of CDH1. However, these cells remained sensitive to the FAK inhibitor PND-1186, regardless of their level of CDH1 mRNA. When analysing the CDH1 expression levels and the correlated sensitivity to PND-1186, we found that this relationship was present only in MPM cells and not in other cancers.

We performed NATA-accredited IHC staining for the E-cadherin protein expression in MPM samples, and did not find significant evidence of its expression (Figure 1). This lack of detection by IHC and the useful detection of CDH1 mRNA results discussed above raise the question regarding when the technique of RT-qPCR should be used instead of IHC staining in order to predict sensitivity to PND-1186 (Figure 2A,C). Schulz et al., in a study of breast cancer, showed that mRNA may not always correlate to the protein expression [20]. In Schulz’s paper, they carried out a single cell analysis and suggested that this discrepancy was due to there being different types of cells in the sample. They found a strong correlation between the mRNA and protein expression at the single cell level, but not at the cell population level. They showed that one of the genes (CXCL10) was probably expressed only by T-cells in the sample, and not by all cells in the sample. One possible explanation for our findings is that E-cadherin is expressed only by fast-growing mesothelioma cells, and that this is detectable through the presence of mRNA, but is not detectable when looking at the whole tissue using IHC. The lack of protein expression could also be due to the E-cadherin antibody binding to a different epitope of the protein that is not present in the samples. An epigenetic alteration mechanism could also contribute to the suppression of the E-cadherin protein expression [21]. To the best of our knowledge, our study is the first to show that the mRNA expression levels of E-cadherin (CDH1) are useful for predicting sensitivity to PND-1186 (Figure 2). The COMMAND trial of another class of FAK inhibitor (VS-6063) failed to show a positive response to FAK inhibitors in MPM [22]. The COMMAND trial used a different indicative biomarker, Merlin, than the one used in our study. Our results suggest that the utility of PND-1186 in treating MPM should be explored further. We suggest that a comprehensive pre-clinical (animal) study should be carried out to further investigate the utility of FAK inhibitors in treating MPM, as well as the effects of E-cadherin mRNA and the protein expression levels on that efficacy.

Our previous studies showed that microRNA has the potential to down-regulate the protein biomarker [23], and the restoration of microRNA led to MPM cell death [24]. In this study, we showed that microRNAs linked to EGFR 3′UTR play a role in MPM cells’ response to FAK inhibitor treatment. More specifically, all candidate microRNAs were found to induce H28 sensitivity to PND-1186 treatment. MicroRNA also exhibited efficacy in the other tested cells (MSTO-211H, Ren, and 1157), although not as pronounced as the response in H28 cells (Figure 5C). Our study suggested that microRNA plays a role in sensitising PND-1186 in MPM cells. The current study suggests that a more detailed prospective study to investigate the 3′UTR interaction with different microRNAs is warranted.

## 4. Materials and Methods

### 4.1. Patient Samples and Cell Lines

All of the samples of this project were approved by the Human Research Ethics Committees at Concord Repatriation General Hospital. This study includes 82 MPM FFPE tissues and 18 cell lines, including American Type Culture Collection (ATCC) cell lines and ADRI-established primary MPM cell lines (Appendix A). The human MPM cell lines H28 and MSTO-211H, and the immortalised human mesothelial cell line, MeT-5A, were purchased from the ATCC (Manassas, VA, USA). Ren human mesothelioma cells were provided by Laura Moro (University of Piemonte Orientale A. Avogadro, Novara, Italy). The primary human mesothelioma cell line MM05 was generated at the University of Queensland Thoracic Research Centre (The Prince Charles Hospital, Brisbane, Australia) [25]. The VMC23 cell line was kindly provided by A/Prof Michael Grush (Medical University of Vienna). Primary non-cancer human dermal fibroblasts (Humofib) were also included. Details of the cell types are included in Appendix A. All cell lines were cultured in an RPMI-1640 medium with 10% foetal bovine serum (FBS), and were maintained at 5% CO_2_, 37 °C, and 95% humidity.

### 4.2. MPM Cell Block

ATCC and MPM primary cells were harvested and fixed in buffered formalin and were embedded into cell blocks that were further processed into paraffin blocks. MPM cell blocks were sectioned at 3 μm thickness and processed for immunohistochemistry (IHC) staining.

### 4.3. Immunohistochemistry

Tissue sections were deparaffinised and rehydrated in graded concentrations of xylene and ethanol. Antigen retrieval and IHC staining were performed on an automated Leica Bond III (Leica Microsystems, Macquarie Park, NSW 2113, Australia) using a Bond Polymer Refine Detection Kit (Leica Biosystems, Newcastle upon Tyne, NE12 8EW, UK). Heat induced epitope retrieval (HIER) was performed on all slides in a Bond Epitope Retrieval Solution (Leica Biosystems, Newcastle upon Tyne, NE12 8EW, UK) 2 (pH9) for 20 min. Primary E-cadherin antibody (Abcam, Cat: ab1416) was applied and incubated for 20 min at room temperature. The slides were then immersed in H_2_O_2_ for 5 min to quench the endogenous peroxidases. The slides were processed for post-primary detection for 15 min, followed by a polymer for 15 min. 3,3′-diaminobenzidine (DAB with enhancer) chromogenic detection and haematoxylin counterstaining were used. Diagnostic clinical procedures related to the diagnosis of the cases were performed in our NATA-approved laboratory using external quality assurance program (QAP) validated tests. The method of scoring for each antibody in each case was as per clinical diagnostic practice. A negative staining pattern was defined as no staining. Positively stained cells were defined as 1+ (weak), 2+ (moderate), or 3+ (strong) staining intensity in the cells, and the number of cells showing the relevant positive intensity was scored as a percentage over the total number of cells present.

### 4.4. RNA Isolation and RT-qPCR

Quantitative reverse transcription PCR (RT-qPCR) was carried out following the protocol published in our previous study [25]. Briefly, the total RNA was extracted from the cell lines using a Trizol reagent (Life Technologies, Thornton, NSW 2322, Australia) following the manufacturer’s protocol. Reverse transcription reactions were performed using 200 ng of total RNA with a MMLV first strand cDNA kit (Promega, Madison, WI, USA), following the manufacturer’s protocol. The gene expression was determined by RT-qPCR using the KAPA SYBR^®^ FAST qPCR Master Mix (Sigma-Aldrich, Castle Hill, NSW 2154, Australia) and the Vii7 QPCR System (Thermo Fisher Scientific, North Ryde, NSW 2113,Australia). Probe Design software (Roche Diagnostic, North Ryde, NSW 2133, Australia) was used for designing the PCR primers (Appendix A). For each gene in a cell line, the qPCR cycle threshold was collected for three replicates. The expression levels of the mRNA were determined using the 2^−∆∆Cq^ method [26] with normalisation to the reference gene.

### 4.5. Drug Treatment and AlamarBlue^®^ Proliferation Assay

AlamarBlue^®^ cell death assays were carried out for the cells as follows. The cells were plated in 96-well culture plates at 2500 cells in 100 μL medium per well for the drug response study. For the microRNA study, before plating the cells, we transfected the cells with candidate microRNAs (miR-17, miR221, miR-222, miR137, and miR148), as described previously [27], and after 24 h, the cells were treated with PND-1186 at various concentrations, and then the incubation of cells continued for 72 h. Then, 15 μL AlamarBlue^®^ (50 mL PBS containing also Sigma reagents 0.075 g Resazurin, 0.0125 g Methylene Blue, 0.1655 g Potassium hexacyanoferrate (III), and 0.211 g Potassium hexacyanoferrate (II) trihydrate, filter-sterilised, and stored at 4 °C in the dark) was added and incubated for 4 h at 37 °C. The fluorescence intensity was measured at 590.10 nm with 544 nm excitation, using a FLUOstar Optima (BMG LabTech, Ortenberg, Germany). The fluorescence intensity was calculated as a percentage of the intensity of the control cells (untreated). Experiments involving human cell lines were performed three times and were replicated three times each time. All of the media and FBS were from Life Technologies (Carlsbad, CA, USA).

### 4.6. Cell Cycle Analysis

We selected four cell lines, MSTO-211H, H28, Ren, and 1157, for the cell cycle analysis after treating with PND-1186. At 72 h post-PND-1186 treatment, the cells were harvested and washed three times with phosphate-buffered saline (PBS). The cells were fixed with 70% ethanol for at least 30 min at room temperature and were subject to cell cycle analysis. For the cell cycle analysis, the fixing solution was removed and the cells were treated with 0.01% RNase (10 mg/mL, Sigma) and 0.05% PI in PBS for 30 min at 37 °C in the dark. The cell cycle distribution was determined on a CytoFLEX flow cytometer (Beckman) within 30 min. The flow cytometer was calibrated using calibration beads provided by the manufacturer, and according to their instructions (CYtoFLEX, Beckman). The flow cytometer was routinely operated at the Slow Flow Rate setting (14 μL sample/minute), and the data acquisition for a single sample typically took 3–5 min. For each sample, 10,000 events of single cells were counted and the cell cycle was analysed using FlowJo software (Ashland, OR, USA).

### 4.7. Statistical Analysis and Protein-Protein Interaction Analysis

Statistical analyses were carried out using R installation on IBM SPSS Statistics 25 software and on Linux [28]. IC50 values were calculated with R package “drc” using the LL3 three-parameter logistic function for correlating the results with the mRNA fold change values. Distributions were normal but had a sparsity of points, and thus Spearman’s rank correlation was used to calculate the correlations between mRNA fold change and IC50 values. R’s loess function was used to display the correlation for plotting. R’s lm linear regression module was used to calculate the linear regression and to display the correlation for plotting. A protein−protein interaction analysis was performed using String software at String website https://string-db.org/ assessed on 9 August 2021 [29] website functionality at https://string-db.org/ assessed on 9 August 2021. Scanning for human microRNA associated with human genes was carried out using TargetScanHuman software at website http://www.targetscan.org/vert_72/ assessed on 9 August 2021 [30].

## 5. Conclusions

Our observations in this study indicate that the presence of CDH1 mRNA in MPM cells is associated with resistance to the FAK inhibitor (PND-1186), regardless of E-cadherin protein expression measurements by IHC. Our results also indicate that microRNAs are able to sensitise FAK-inhibitor-resistant MPM cells to FAK inhibitor-induced cell death.

## Figures and Tables

**Figure 1 ijms-22-10225-f001:**
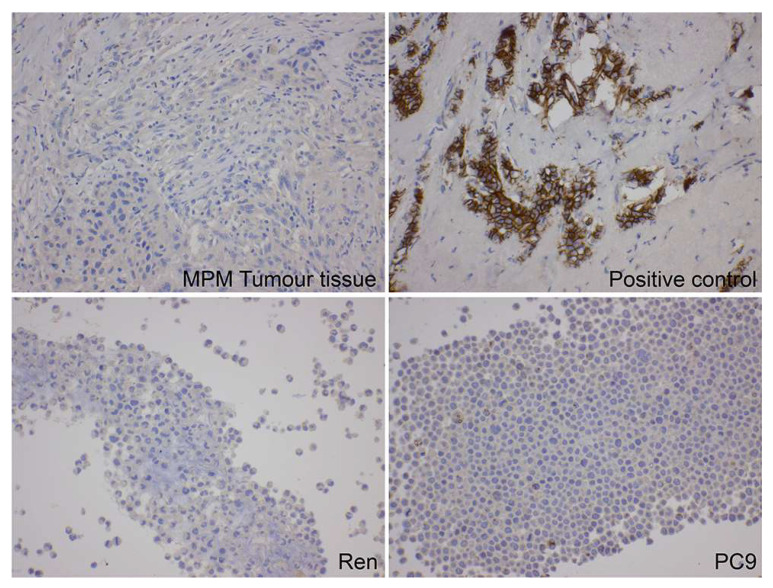
Protein expression as measured by IHC of E-cadherin for representative samples of the FFPE MPM tumour (**top left**), MPM cell line Ren (**bottom left**), lung cancer cell line PC9 (**bottom right**), and positive control breast tissue sample (**top right**). Images were captured with a ZEISS Axio.M2 microscope with 20× objective. Most MPM tumour and cell lines exhibited E-cadherin protein silencing.

**Figure 2 ijms-22-10225-f002:**
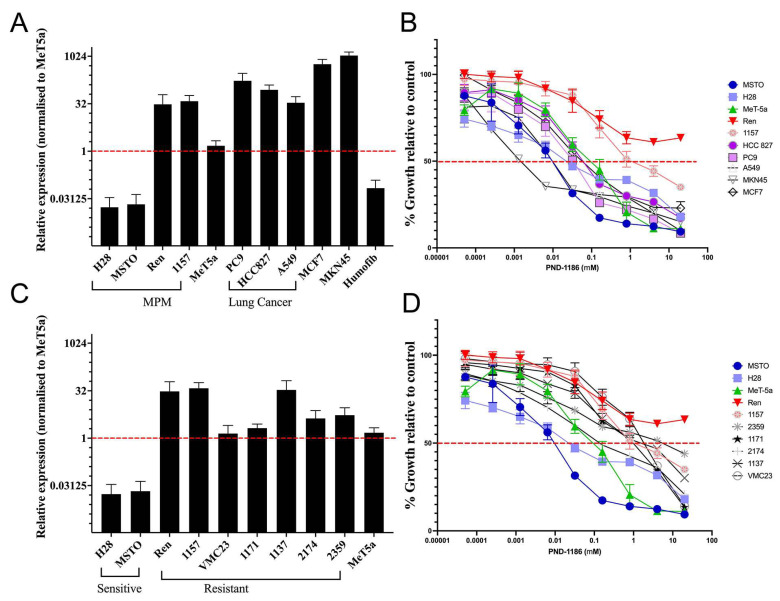
(**A**,**C**) Relative mRNA expression of E-cadherin (CDH1) in (**A**) mesothelioma and non-MPM cancer cell lines and (**C**) mesothelioma cell lines, showing the mRNA expression relative to the immortalised non-cancer mesothelial cell line MeT-5a. The mRNA expression was analysed using Thermo Fisher Vii7 qPCR, normalised to reference gene 18s. (**B**,**D**) FAK inhibitor PND-1186 response curve in (**B**) mesothelioma and non-MPM cancer cell lines and (**D**) only mesothelioma cell lines. The horizontal red line at 50% viability indicates the IC50 concentration for each cell line, at the point that the dose response curves cross the line.

**Figure 3 ijms-22-10225-f003:**
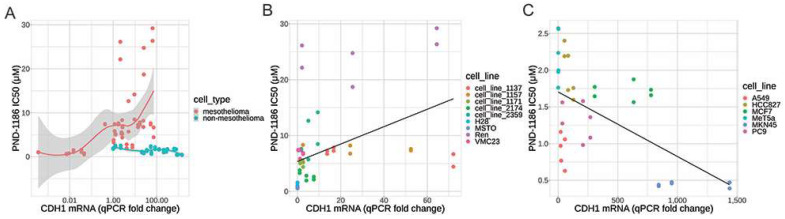
Plots of the correlation between the CHD1 mRNA fold change levels, as measured by RT-qPCR, and the PND-1186 IC50 drug response values. (**A**) Loess regression line fitting mRNA to IC50 for mesothelioma cell lines (orange) and non-MPM cell lines (cyan). (**B**) Linear regression line (black) fitting mRNA to IC50 for mesothelioma cell lines. (**C**) Linear regression line (black) fitting mRNA to IC50 for non-MPM cell lines.

**Figure 4 ijms-22-10225-f004:**
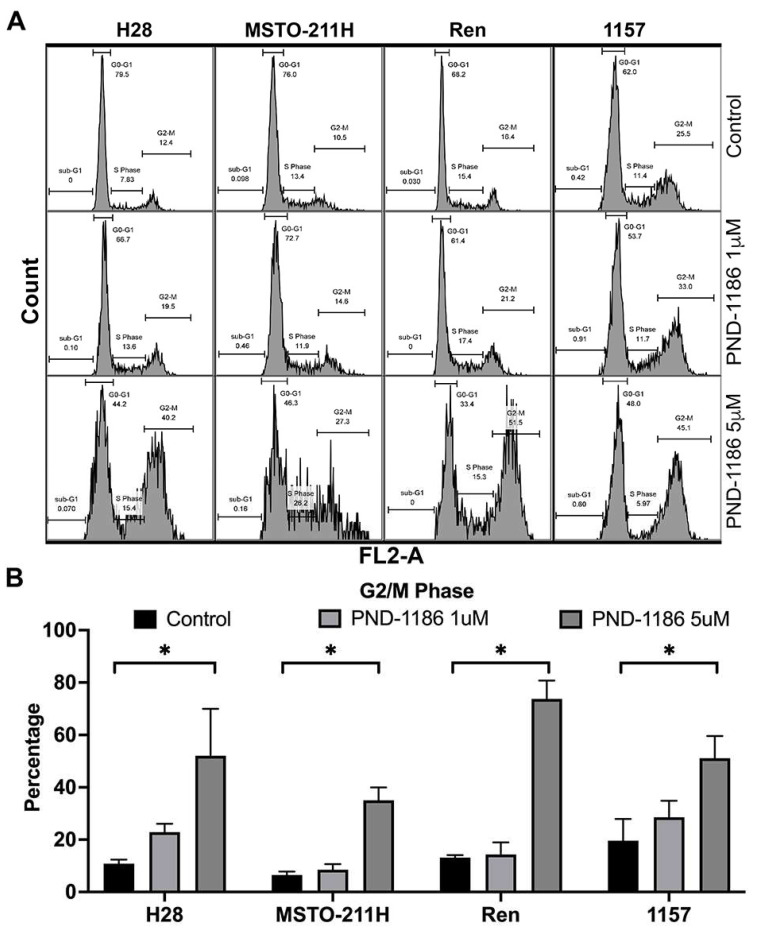
Cell cycle analyses were performed using a CytoFLEX (Beckman) instrument for cells treated with PND-1186 at 72 h post-treatment at the concentrations specified on the graph. (**A**) Representative cell cycle analysis on different phases are shown. (**B**) The percentages of the G2/M phase for each cell line are plotted, and statistically significant differences with respect to the untreated control are indicated with *.

**Figure 5 ijms-22-10225-f005:**
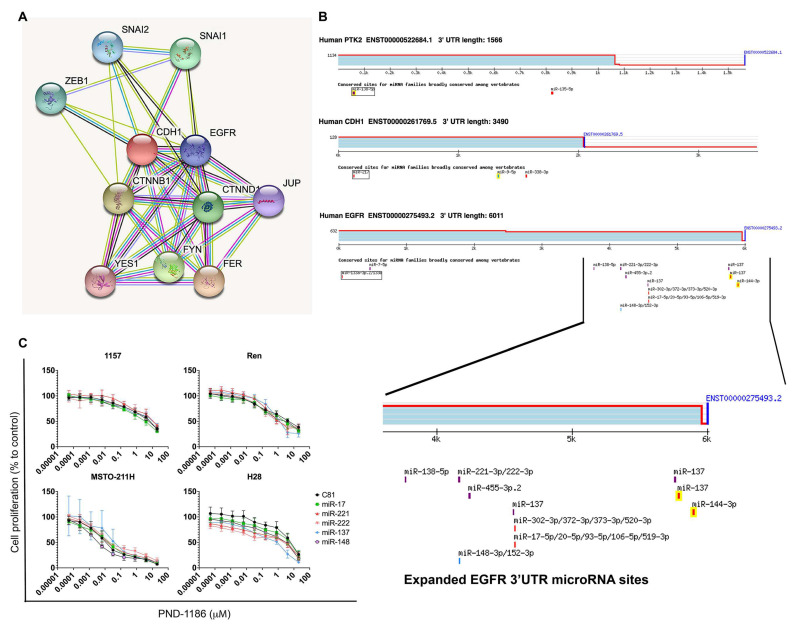
(**A**) String protein−protein interaction result showing proteins interacting with CDH1 (E-cadherin) (**B**) TargetScanHuman results of microRNA associated with genes that possibly affect the FAK pathway and its inhibitor. (**C**) Plots showing that microRNAs sensitised MPM cells to PND-1186. MPM cell lines were transfected with candidate microRNAs (miR-17, miR-221, miR-222, miR-137, and miR-148). At 24 h post transfection, the cells were treated with increasing concentrations of PND-1186 and proliferation was determined at 72 h. The controls received no microRNA. The results showed that microRNA-treated cells exhibited less cell proliferation (more cells were killed by the PND-1186 treatment) than the controls.

**Table 1 ijms-22-10225-t001:** E-cadherin IHC average results for 81 MPM FFPE clinical samples.

Mesothelioma Subtype	E-Cad IHC 2+ (%)	E-Cad IHC 1+ (%)	E-Cad IHC Negative (%)
Biphasic	4% (1/28)	25% (7/28)	71% (20/28)
Epithelioid	8% (3/37)	35% (13/37)	57% (21/37)
Sarcomatoid	0% (0/16)	6% (1/16)	94% (15/16)

**Table 2 ijms-22-10225-t002:** PND-1186 IC50 of all of the cell lines tested.

Cell line	IC50 (µM)
MSTO	0.70 ± 0.05
H28	1.20 ± 0.12
MeT-5A	2.18 ± 0.16
Ren	24.86 ± 3.91
1157	7.20 ± 0.52
HCC827	1.96 ± 0.15
PC9	1.28 ± 0.10
A549	0.89 ± 0.11
MNK45	0.44 ± 0.05
MCF7	1.71 ± 0.12
2359	9.16 ± 0.90
1171	5.11 ± 0.17
2174	2.71 ± 0.16
1137	6.63 ± 0.35

## Data Availability

All data of this article either included in the main text or Appendix A.

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
