# Peer review of "The Role of E-Cadherin and microRNA on FAK Inhibitor Response in Malignant Pleural Mesothelioma (MPM)"

_ijms, 2021, doi:10.3390/ijms221910225_

Round 1

Reviewer 1 Report

This is an interesting and well-writing manuscript. The subject is topical and hypotheses are original. In summary, the results showed that MPM cells that were relatively resistant to FAK inhibitor (PND-1186) showed high CDH1 mRNA levels, regardless of E-cadherin protein expression measurements by IHC. PND-1186 induced cell cycle disruption and the authors found possible microRNAs (miR-17, miR221, miR-222, miR137 and 31 miR148) able to sensitize FAK-inhibitor-resistant MPM cells to FAK inhibitor-induced cell death. The discussion section is well described.

This paper is interesting and suitable with the concern and purpose of the journal and I advice the publication.

Author Response

We thank the reviewer's valuable comments, we have prepared an updated version with track changes.

Reviewer 2 Report

The manuscript titled "The role of E-cadherin and microRNA on FAK inhibitor response in malignant pleural mesothelioma (MPM)" by Yuen Et al. provides convincing data on the impact of E-cadherin expression in the responses of mesothelioma cells on the FAK inhibitor PND-1186. The study is well-designed and the results support the conclusions. I have the following (minor) suggestions:

1) PND-1186 should be adressed as the inhibitor used in this study in the abstract (in the first instead of the second sentence that it appears) and in the last sentence of introduction.

2) The mentioned results should be appear more clearly in the discussion section, using references to respective figures. 

3) The alterations in the cell cycle are not discussed. I believe that a couple of sentences should be added to comment on this interesting phenomenon. 

Author Response

The manuscript titled "The role of E-cadherin and microRNA on FAK inhibitor response in malignant pleural mesothelioma (MPM)" by Yuen Et al. provides convincing data on the impact of E-cadherin expression in the responses of mesothelioma cells on the FAK inhibitor PND-1186. The study is well-designed and the results support the conclusions. I have the following (minor) suggestions:

Point 1. PND-1186 should be addressed as the inhibitor used in this study in the abstract (in the first instead of the second sentence that it appears) and in the last sentence of introduction.

Response 1:

Original sentences: We showed that MPM cells having high CDH1 mRNA levels exhibited resistance to PND-1186. In summary, MPM cells that did not express CDH1 mRNA were sensitive to the PND-1186 FAK inhibitor, and MPM cells that retained CDH1 mRNA were resistant. Cell cycle analysis showed that PND-1186 induced cell cycle disruption by inducing G2/M arrest of MPM cells.

With track changes included we have followed the Reviewer’s suggestion and changed the above to:

We showed that MPM cells having high CDH1 mRNA levels exhibited resistance to PND-1186 (a FAK inhibitor). In summary, MPM cells that did not express CDH1 mRNA were sensitive to the PND-1186, and MPM cells that retained CDH1 mRNA were resistant. Cell cycle analysis showed that PND-1186 induced cell cycle disruption by inducing G2/M arrest of MPM cells.

Point 2. The mentioned results should be appear more clearly in the discussion section, using references to respective figures. 

Response 1: As recommended by the reviewer, we have cited our result figures in the discussion section with tracked changes.

Point 3. The alterations in the cell cycle are not discussed. I believe that a couple of sentences should be added to comment on this interesting phenomenon. 

Response 1: In order to follow the suggestions of the Reiveres, in the second paragraph of the discussion, we have included additional discussion as below with additional reference 18 and 19.

The striking correlation between CHD1 mRNA expression and resistance to cell death by PND-1186 treatment seen in MPM cells was not seen in non-MPM cells (Figure 3).

and

There is limited study reviewing the PND-1186 induced cell cycle alterations in mesothelioma. Tancioni et al., suggested G0/G1 alterations in breast cancer (18) whiles Tanjoni et al suggested no cell cycle alterations in 4T1 cells (19). We performed cell cycle analysis on two PND-1186 sensitive lines (H28 and MSTO-211H)) and two resistant lines (Ren and 1157). Our result indicated that all MPM cell lines showed some degree of cell cycle alteration at low dose (1mM) and at high dose (5mM) all cells showed G0/G1 as well as G2/M alterations. We showed in our study that PND-1186 is able to induce cell cycle alterations in MPM cells regardless of its sensitivity potential.

  1. Tancioni I, Miller NL, Uryu S, Lawson C, Jean C, Chen XL, et al. FAK activity protects nucleostemin in facilitating breast cancer spheroid and tumor growth. Breast Cancer Res. 2015;17:47.
  2. Tanjoni I, Walsh C, Uryu S, Tomar A, Nam JO, Mielgo A, et al. PND-1186 FAK inhibitor selectively promotes tumor cell apoptosis in three-dimensional environments. Cancer Biol Ther. 2010;9(10):764-77.